# Continuum Electronic States: The Tiresia Code

**DOI:** 10.3390/molecules27062026

**Published:** 2022-03-21

**Authors:** Piero Decleva, Mauro Stener, Daniele Toffoli

**Affiliations:** 1Istituto Officina dei Materiali IOM-CNR and Dipartimento di Scienze Chimiche e Farmaceutiche, Università degli Studi di Trieste, I-34121 Trieste, Italy; stener@units.it; 2Dipartimento di Scienze Chimiche e Farmaceutiche, Università degli Studi di Trieste, I-34121 Trieste, Italy; toffoli@units.it

**Keywords:** molecular photoionization, electronic continuum states, theoretical chemistry

## Abstract

A multicenter (LCAO) B-spline basis is described in detail, and its capabilities concerning affording convergent solutions for electronic continuum states and wavepacket propagation are presented. It forms the core of the Tiresia code, which implements static-DFT and TDDFT hamiltonians, as well as single channel Dyson-DFT and Dyson-TDDFT descriptions to include correlation in the bound states. Together they afford accurate and computationally efficient descriptions of photoionization properties of complex systems, both in the single photon and strong field environments. A number of examples are provided.

## 1. Introduction

Quantum Chemistry (QC) approaches to the calculation of electronic bound states of complex molecules have reached a large degree of sophistication, and have given rise to a multitude of highly advanced computer codes. The treatment of continuum states to describe photoionization and electron scattering has significantly lagged behind, probably because of unfamiliarity with continuum states in the chemistry community, the unsuitability of bound state algorithms for the determination of approximate solutions, and the limitations of the ubiquitous Gaussian type orbital (GTO) basis sets in describing continuous states. A strong revival of interest in molecular continuum states, especially for ionization, has been spurred by the development of novel light sources, high intensity laser and Free Electron Lasers (FELs), capable of ultrashort pulses, together with the continuous development of Synchrotron Radiation (SR) sources, and the parallel development of ever more performant detectors for electrons and ions, which have produced a flurry of new experiments affording unprecedented detail in studies of light–matter interaction in molecules [1].

In the treatment of continuum states, two new problems appear. The first is the accurate solution of the one-particle Schrödinger equation for positive energies. The second is the treatment of many-electron systems, compatible with the asymptotic structure of the wavefunction. Initial studies on atoms naturally employed numerical radial grids. While this approach can be generalized to a one center expansion (OCE) of the wavefunction in the molecular case [2,3], it suffers from the very slow convergence of the partial wave expansion for heavy atoms far from the expansion center. Generalization to 3D grids is difficult, because a natural coordinate system is spherical around each nucleus, due to the Coulomb singularity and the relative cusps in the wavefunction. In the Continuum Multiple-Scattering Method (CMS) shape approximation of the potential allows efficient numerical integration [4]. Other adaptive grids have been suggested [5]. Cartesian grids are, however, in use, employing pseudopotentials to smooth the Coulomb singularities, especially for the treatment of time dependent phenomena [6]. Another difficulty is the implementation of asymptotic boundary conditions, which are most neatly expressed in a partial wave expansion. The obvious solution is suggested by the well established LCAO approach in molecules. The superposition of spherical basis sets centered on the various atoms effectively mimics the change of coordinates from spherical around one site to spherical around another, dealing very effectively with coulomb singularities. In the continuum case these have to be added to the OCE expansion, which is needed to describe the long range part of the wavefunction. This prompts the use of a basis set approach, requiring, however, new basis functions, capable of approximating the oscillatory behavior of continuum wavefunctions. Note however that the expansion can be limited to a finite spherical volume, large enough that all bound states of interest are negligible beyond this boundary, and the continuum states have reached their asymptotic form, so that they can be fitted to analytical forms (spherical Bessel or Coulomb functions), or to the solutions of a weak multipolar field by an OCE approach (Figure 1).

Recent work has converged to the use of local polynomial functions as radial basis, finite elements [7] or B-splines [8,9,10], times spherical harmonics, at least for the long range OCE part, although GTOs have been used for small ranges and low electron kinetic energies [9]. For the short range, multicenter part, often GTOs are used, in various combinations [9,10].

The second issue is the treatment of the many electron problem. A single determinant description of the bound states is easily extended to the continuum, replacing a bound state orbital with a continuum one, obtained by solving a resultant one particle equation, like Frozen Hartree–Fock (HF) or Density Functional (DFT). If $|\Phi_{0}\rangle$ denotes the HF or DFT ground-state of the N-electron system:(1)$$\Phi_{0}=|\phi_{1}\cdots \phi_{i}\cdots \phi_{N}\rangle ,$$
then the set of degenerate solutions at energy E>0 can be written as:(2)$$\Phi_{Eilm}=|\phi_{1}\cdots \phi_{Elm}\cdots \phi_{N}\rangle ,$$
where index *i* in Equation (Equation 2) labels the MO from which the electron is ejected. In HF theory, the approach is called Static Exchange (SE), while it has been termed SE-DFT in DFT theory. Coupling among channels is implemented as Configuration Interaction Singles (CIS), also known as the Tamm–Dancoff approximation (TDA). Some initial state correlation can be included at the linear response level, giving the well known random phase approximation (RPA) or equivalent Time Dependent DFT (TDDFT), in the ab initio or DFT framework, respectively. This can be generalized to the use of correlated bound states, both initial and final, giving rise to the so called close-coupling form of the wavefunction
(3)$$\Psi_{EIlm}^{N}=\sum_{J}\Psi_{J}^{N-1}\phi_{EIJlm}+\sum_{K}\Phi_{K}^{N}C_{KI}.$$

In Equation (Equation 3), the first term comprises antisymmetrized products of N-1 particle bound eigenstates $\Psi_{J}^{N-1}$, describing the final states of interest (target states), times continuum orbitals $\phi_{EIJlm}$, and ensures the correct asymptotic limit. The second part describes short range correlations and autoionization resonances. Actually, some restrictions have to be included to avoid linear dependencies and ensure unique solutions [11]. Given its generality, it can be considered as the equivalent of CI in the continuum, and several variants can be designed depending on specific forms for the bound states included in the expansion. For instance, restricting the target states to single hole states relative to an initial closed shell state, and ignoring correlating functions, one recovers the CIS ansatz, or by including a single, correlated target state $\Psi_{I}^{N-1}$, relative to the final state of interest, one has the Single Channel (SC) approximation.
(4)$$\Psi_{EIlm}^{N}=\Psi_{I}^{N-1}\phi_{EIlm}.$$

At this level, correlation is fully included in the bound states, but neglected in the continuum. The close-coupling expansion can in principle be very accurate, but it is computationally expensive, and simpler approximations are often employed.

## 2. The Tiresia Program

### 2.1. The Basis Set

Our current implementation uses B-splines for the OCE and also for the LCAO part. B-splines (B is for basis) are functions defined over a set of points (knots) which divide a given interval, $\left[0,R_{max}\right]$ in our case, in $n_{int}$ subintervals. Over each subinterval a B-spline is a different polynomial of a common fixed order *n* (degree n−1), spanning n+1 consecutive knots, and is zero outside. Different polynomial pieces are joined at the boundaries with a high degree of continuity, typically the maximum admissible, i.e., $C^{\left(n-2\right)}$, continuity up to the derivative of order n−2, but that can be relaxed employing coincident knots. In fact in the present implementation continuity is completely relaxed at the endpoints, 0 and $R_{max}$, putting here *n* coincident knots, so that the first and last spline are nonzero there, while employing single knots, i.e., maximum continuity, at the interior points. This ensures a complete basis, entirely contained in the interval $\left[0,R_{max}\right]$. Actually the first spline is deleted, so as to enforce the boundary condition f(0)=0. The same is done at $R_{max}$ when dealing with bound states, assumed to be negligible at the outer boundary. The last spline is instead retained in continuum calculations, as the solution naturally oscillates and is nonzero at the boundary. With this choice the dimension of the basis is $n+n_{int}-1$, so the B-spline order has a very limited impact on the dimension, but higher orders vastly increase accuracy, before degrading due to numerical truncation errors. B-splines as well as their derivatives are easily evaluated at a given point, and integration by Gauss–Legendre quadrature is accurate down to machine precision. A full discussion including fortran routines is available in the monograph [12], and a detailed treatment of atomic and molecular applications can be found in [13].

In the present case the B-spline basis is completely defined by the polynomial order *n* (default value is *n* = 10) and the knot sequences that define intervals $\left[0,R_{max}\right]$ and $\left[0,R_{maxi}\right]$ of the OCE and LCAO expansions around each atom. In the OCE sequence it is important to have a knot at the radial position of each atom, as the singularity of the potential there degrades the accuracy of the numerical integration otherwise. As the primitive functions are
(5)$$\chi_{ilm}\left(r,\theta ,\phi \right)=\frac{1}{r}B_{i}\left(r\right)Y_{lm}\left(\theta ,\phi \right)$$
the complete basis is specified by the maximum *L* values of the spherical harmonics for the OCE, $L_{max}$ and the LCAO part, $L_{max,i}$. The OCE part is numerically very stable; in normal photoionization typical $R_{max}$ around 20–40 atomic units (au) are employed, but values up to several thousands have been employed for long wavepacket propagation without trouble. For photoionization a linear grid of knots is adequate, with spacing *h* being determined by the maximum electron momentum $k_{max}$ (electron kinetic energy $KE=k_{max}^{2}/2$). A choice $h=1/k_{max}$ is generally very accurate. $L_{max}$ has to accommodate both convergence of the expansion in the molecular region, and the number of continuum partial waves carrying significant contribution at $k_{max}$. For the latter, $L_{max}$ is roughly proportional to $k_{max}$ times the “size of the molecule”, by the classical argument based of momentum times impact parameter. So large systems and high electron energies may require large $L_{max}$, and small step *h*, making the basis very large. In the case of heavy atoms, steps have to be reduced close to the origin. If only bound states are needed, e.g., high Rydberg states, an exponential grid becomes more convenient. Then, with appropriate range, quantum numbers above 100 can be reached.

In contrast, LCAO $R_{max,i}$ are generally very small, of the order of about 1 au. As corresponding $L_{max,i}$ are the usual values of the maximum *L* in the free atom, or L+1, the dimension of the LCAO basis is generally very small compared to OCE, which alone determines the computational cost of the calculation. The very short range of the LCAO expansion is dictated by the need to avoid overcompleteness, i.e., numerical linear dependence of the basis, which destroys numerical stability. Actually, the overcompleteness problem is even more severe with GTOs, which have a very large overlap. The ease of selecting the range of B-splines makes it easier to control overcompleteness. There is actually a bargain between LCAO radii and OCE $L_{max}$, as with larger $L_{max}$ shorter $R_{max,i}$ have to be employed, and with low $L_{max}$ larger $R_{max,i}$ make up for convergence in the molecular region. Limiting the LCAO basis to non-overlapping ranges in the present implementation does not constitute a limitation and has the additional benefit of reducing the number of matrix elements to be evaluated, as products of LCAO functions on different centers are strictly zero, $B_{i}\left(r_{p}\right)B_{j}\left(r_{q}\right)=0$. The last LCAO splines (usually 3) are then deleted, so that the basis has overall continuous second derivative at all internal points. All two center integrals between OCE and LCAO functions are accurately evaluated by numerical integration within the LCAO spheres.

The basis is then fully adapted to point group symmetry, if present, which brings a great reduction in the dimension of the matrices and the computational effort
(6)$$\chi_{ij\lambda \mu }\equiv \chi_{ilh\lambda \mu }=\frac{1}{r}B_{i}\left(r\right)\sum_{m}Y_{lm}\left(\theta ,\phi \right)b_{lmh\lambda \mu }$$
for the OCE basis, and
(7)$$\chi_{pij\lambda \mu }=\sum_{q\in p}\frac{1}{r_{q}}B_{i}\left(r_{q}\right)\sum_{m}Y_{lm}\left(\theta_{q},\phi_{q}\right)b_{qlmh\lambda \mu }$$
for the LCAO part. Here (λμ) are the labels for the irreducible representations, j≡lh counts the number of total angular independent components, with *h* counting the independent components relative to a given *l* value. For the LCAO functions, *p* counts the number of sets (shells) of symmetry equivalent atoms, and *q* runs over the atoms in this set. The final basis is then
(8)$$\chi_{\nu }=\left\{\chi_{ij\lambda \mu }\right\}\cup_{p}\left\{\chi_{pij\lambda \mu }\right\}$$

The final basis has a lot of flexibility, and can afford good convergence for a rather wide range of parameters. It requires, however, some care, as it is easy to overdo, which is signalled by the minimum eigenvalues of the overlap matrix *S* that become very low. It is then necessary either to restrict the range of LCAO functions or decrease the OCE $L_{max}$. The upside is that it is possible to accomodate a large variety of situations.

### 2.2. The Galerkin Approach

The one particle Schrödinger equation:(9)hϕ=Eϕ
is converted to the generalized eigenvalue problem
(10)hc=ESc,
when the solution ϕ is expanded as a linear combination of basis functions
(11)$$\phi =\sum_{\nu }\chi_{\nu }c_{\nu }$$

In Equation (Equation 10) *h* and *S* denote the Hamiltonian and overlap matrices, respectively, of dimension equal to the number of basis functions employed in the expansion of ϕ. Bound state eigenvectors are obtained from a usual diagonalization. It is important to have accurate initial orbitals, in particular in the outer tails, which are often not well described by conventional basis sets, like GTO, because the transition matrix elements for the continuum prove sensitive to them. In the case of continuum eigenvectors several algorithms can be used. Note that in this case, because of the last nonzero spline at the end of the interval, the Hamiltonian matrix, *h*, is non-Hermitian. We employ the Galerkin approach, originally proposed in [14], which amounts to requiring orthogonality of the residual vector (h−E)ϕ to all basis functions. Defining the energy dependent matrix A(E)=h−ES, this amounts to solving:(12)A(E)c=ac
for the $n_{o}$ lowest eigenvalues, where $n_{o}$ is the number of open channels, i.e., the number of independent solutions. Actually, an alternative formulation [15,16], employing the hermitian product $A^{T}A$ is more stable and accurate. It turns out that, with a good basis, the gap between the lowest $n_{o}$ eigenvalues and the following ones is many orders of magnitude, and the eigenvectors are easily extracted with a block inverse iteration, which generally converges very fast. The resulting continuum eigenvectors are fitted to a linear combination of regular and irregular asymptotic functions and are then normalized to the K-matrix asymptotic form. Full details are in [15]. Note that, as the basis is close to complete within the sphere, any type of linear equation, either homogenous or inhomogeneous can be accurately solved in matrix form. We currently employ this technique to evaluate the Coulomb potential generated by a charge distribution solving the Poisson equation, to solve inhomogenous equations in TDDFT and so on.

### 2.3. The Many Electron Wavefunction

#### 2.3.1. The Static Exchange Approach

The Static Exchange (SE) approach is defined by the use of the same (frozen) orbitals for the ground state as for for the ionic states. The simplest approach is the DFT-SE scheme. We employ a fixed Kohn–Sham (KS), or DFT, single particle hamiltonian
(13)$$h_{KS}=-\frac{1}{2}▵+V_{N}\left(r\right)+V_{C}\left(r\right)+V_{XC}\left(r\right)$$
where $V_{N}$ is the nuclear attraction potential, $V_{C}$ is the classical Coulomb (Hartree) potential generated by the electronic density ρ(r) and $V_{XC}\left[\rho \left(r\right)\right]$ is a local exchange-correlation potential defined by the same density. The density is that of the ground state (GS), obtained by a previous self-consistent field (SCF) calculation with a separate QC program. The *h* and *S* matrices are computed in the B-spline basis, and the occupied orbitals obtained by diagonalization, defining the initial state in Equation (Equation 2). At each energy E>0, $n_{o}$ independent degenerate solutions are obtained by inverse iteration:(14)$$h_{KS}\phi_{Elm}=E\phi_{Elm}$$
with the asymptotic behaviour (close to $R_{max}$):(15)$$\phi_{Elm}=\sum_{l^{\prime }m^{\prime }}R_{El^{\prime }m^{\prime }lm}\left(r\right)Y_{l^{\prime }m^{\prime }}\left(\theta ,\phi \right).$$

In Equation (Equation 15), the radial functions $R_{El^{\prime }m^{\prime }lm}\left(r\right)$ can be written as:(16)$$R_{El^{\prime }m^{\prime }lm}\left(r\right)=A_{El^{\prime }m^{\prime }lm}\;f_{l^{\prime }}\left(kr\right)+B_{El^{\prime }m^{\prime }lm}\;g_{l^{\prime }}\left(kr\right)$$
where $f_{l}$ and $g_{l}$ are the regular and irregular radial solutions in the asymptotic region, and the matrices *A* and *B* are obtained by fitting the numerical solution to the asymptotic form at the boundary (we use the two last interior knots [14]). Multiplying the solution by $A^{-1}$ gives the final *K*-matrix normalized form:(17)$$R_{El^{\prime }m^{\prime }lm}\left(r\right)=\delta_{ll^{\prime }}\delta_{mm^{\prime }}\;f_{l^{\prime }}\left(kr\right)+K_{l^{\prime }m^{\prime }lm}\left(E\right)\;g_{l^{\prime }}\left(kr\right)$$

From the final state $\Phi_{Eilm}$ (Equation (Equation 2)) the transition dipole matrix elements are evaluated as:(18)$$D_{Eilm\gamma }=\langle \Phi_{Eilm}|D_{\gamma }|\Phi_{0}\rangle =\langle \phi_{Elm}|d_{\gamma }|\phi_{i}\rangle$$
where γ is the dipole operator component. These, together with the *K*-matrix, are passed to a separate program that evaluates photoionization cross sections and angular distributions.

#### 2.3.2. The TDDFT Approach

We employ first-order linear response, which describes the screening of the external exciting field due to the response of the electron cloud. The fundamental equations governing linear response are
(19)$$\delta \rho =\chi V^{\mathrm{SCF}},\;\;\;\;\;\delta V=K\delta \rho ,\;\;\;\;\;V^{\mathrm{SCF}}=\mu^{\mathrm{ext}}+\delta V$$
where δρ is the first-order change in electron density induced by the field, χ is the electric susceptibility, δV is the first-order change in the potential induced by a density change δρ, and *K* is the linear kernel relating the two quantities. The final full potential ($V^{\mathrm{SCF}}$) is the sum of the external perturbing potential ($\mu^{\mathrm{ext}}$) and the response potential (δV).

In the response framework, the effect of electron polarization gives rise to an effective potential, which is the sum of the external dipole field and the electronic response potential. The evaluation of the photoelectron matrix element then boils down to replacing the pure dipole field in Equation (Equation 18) by the effective potential $V^{\mathrm{SCF}}$:(20)$$D_{Eilm\gamma }=\langle \phi_{Elm}|V_{\gamma }^{\mathrm{SCF}}|\phi_{i}\rangle$$

From Equation (Equation 19) one can obtain an equation for $V^{\mathrm{SCF}}$
(21)$$V^{\mathrm{SCF}}=\mu^{\mathrm{ext}}+K\delta \rho =\mu^{\mathrm{ext}}+K\chi V^{\mathrm{SCF}}$$

We then solve directly for the response potential $V^{\mathrm{SCF}}$ as the prime dynamical variable, using a non-iterative numerical algorithm [17]. By representing $V^{\mathrm{SCF}}$ as well as the operators *K* and χ in the B-spline basis, the integro-differential equation, Equation (Equation 21), is converted into a linear algebraic one
(22)$$\left(K\chi -1\right)V^{\mathrm{SCF}}=-\mu^{\mathrm{ext}}.$$

The kernel *K* is the sum of the Coulomb potential and the linearized exchange-correlation response
(23)$$K\left(\bar{r},{\bar{r}}^{\prime }\right)=\frac{1}{|\bar{r}-{\bar{r}}^{\prime }|}+\delta \left(\bar{r}-{\bar{r}}^{\prime }\right)\frac{\delta V_{\mathrm{XC}}}{\delta \rho }$$

The computationally expensive part is the calculation of the KS (non-interacting) linear susceptibility, which is energy dependent, and is computed for each photon energy by solving the inhomogeneous first order perturbative equations.
(24)$$h_{KS}\left(E-\varepsilon_{i}\pm \omega \right)\phi_{i}^{\pm 1}=-QV^{\mathrm{SCF}}\phi_{i}$$
where *Q* in Equation (Equation 24) is a projector over the space of KS virtual orbitals and $\phi_{i}^{\pm 1}$ are first order perturbations of the KS orbitals. The completeness of the basis ensures convergent solutions. Full details are available in Ref. [17].

#### 2.3.3. The Correlated Single Channel Approach

One can improve the treatment of bound state correlation by employing fully correlated initial and final target bound states, computed by ab initio CI or equivalent approaches in a single channel framework, Equation (Equation 4). Here we label $\Psi_{I}^{N}$ the initial state and $\Psi_{F}^{N-1}$ the target state. The SC approximation neglects correlation between the target and the continuum, between different channels (interchannel coupling) and autoionization resonances. However, important correlation effects, especially in the inner ionic states, are fully accounted for. Moreover, it deals equally well with multiconfigurational states, like ionization from excited initial states, and multielectron excitations in the final states.

In this approximation the many-particle transition moment reduces to:(25)$$D_{EFIlm\gamma }=\langle \Psi_{EFlm}^{N}|D_{\gamma }|\Psi_{I}^{N}\rangle =\langle \phi_{EFlm}|d_{\gamma }|\phi_{FI}^{D}\rangle +\langle \phi_{EFlm}|\eta_{FI,\gamma }^{D}\rangle$$
where
(26)$$\phi_{FI}^{D}=\sum_{p}\gamma_{FI,p}\;\phi_{p}\;\;\;\;\;\gamma_{FI,p}=\langle \Psi_{F}^{N-1}|a_{p}|\Psi_{I}^{N}\rangle ,$$
and
(27)$$\eta_{FI,\gamma }^{D}=\sum_{p}\chi_{FI\gamma ,p}\;\phi_{p}\;\;\;\;\;\chi_{FI\gamma ,p}=\langle \Psi_{F}^{N-1}|D_{\gamma }a_{p}|\Psi_{I}^{N}\rangle .$$

$\phi_{FI}^{D}$ and $\eta_{FI}^{D}$ are called the Dyson orbital and the conjugate Dyson orbital, respectively, relative to the two bound states $\Psi_{F}^{N-1}$ and $\Psi_{I}^{N}$. So the full transition matrix element reduces to a single particle transition element between the Dyson and the continuum, plus an overlap term of the continuum with the conjugate Dyson [18]. The latter is, moreover, generally neglected, as it is expected to be very small (it is in fact zero in the SE or TDDFT approaches, as the continuum is rigorously orthogonal to the bound eigenstates of the same hamiltonian).

The continuum relative to a given target state can be computed variationally or can be separately approximated. In the present developments we use the same continuum as in the SE-DFT scheme, which is already quite accurate. We call it the Dyson-DFT approach [19]. In practice this amounts to substituting the Dyson orbital for the bound DFT orbital of the SE approach. Dyson orbitals can be computed by several ab initio approaches. A file containing the details of the ab initio data and Dyson orbitals is provided, and the latter, expressed in GTOs, are then reexpressed in the B-spline basis by projection, which is again very accurate, for the ease of computing transition moments with the final continuum. In the same spirit, one can couple the SC treatment, i.e., the Dyson orbital, to the TDDFT continuum, i.e., substituting in the computation of the transition matrix element the external perturbation $\mu_{\gamma }^{ext}$, i.e., $d_{\gamma }$ with the perturbed one $V_{\gamma }^{SCF}$, giving:(28)$$D_{EFIlm\gamma }=\langle \phi_{EFlm}|V_{\gamma }^{\mathrm{SCF}}|\phi_{FI}^{D}\rangle$$
called the Dyson-TDDFT formulation [20]. This hybrid approach proves quite effective when the additional correlation included in TDDFT proves important.

One important issue is the one-electron effective potential used in the continuum calculation. Our choice for the DFT V${}_{XC}$ potential is the V${}_{LB94}$ potential from [21], which was designed to implement an asymptotic coulomb tail, which proves effective for photoionization [22]. Earlier a transition state density (i.e., half electron removed from the ionized orbital) V${}_{X_{\alpha }}$ [23] or V${}_{VWN}$ potentials [24] were employed. V${}_{LB94}$ proves at least comparable or better, and has the additional advantage that the same potential proves adequate to cover the entire ionization spectrum that is from outer valence to deep core states. The potential can be truncated to an analytical coulomb tail. This is necessary, for instance, with very large $R_{max}$, when V${}_{LB94}$ becomes numerically unstable. Finally, the level of correlation for the treatment of bound states in the Dyson approaches is also important. Despite recent advances, highly correlated ab initio approaches are difficult to afford for relatively large molecules and good basis sets. This is an area where further work is certainly warranted.

## 3. Multiphoton and Strong Field Processes

Let us consider for a moment the differential cross section for photoionization, which is expressed via a transition (dipole) matrix element between the initial and the final continuum (or bound) state
(29)$$\frac{d\sigma }{d\vec{k}}\;\;\to \;\;\langle \Psi_{F\vec{k}}^{\left(-\right)}\left|D\right|\Psi_{I}\rangle =\langle \Psi_{F\vec{k}}^{\left(-\right)}|\Phi_{DI}\rangle$$
which can also be seen as a projection of a “final wavepacket” or “prepared state” $\Phi_{DI}$ onto a field free continuum (or bound) final state, eigenvector of the free hamiltonian. The same logic can be employed for wavepackets relative to more complex processes.

One can consider separately

1.Calculation of $\Phi_{DI}$, which is $L^{2}$ and depends on the particular excitation mechanism considered.2.Calculation of $\Psi_{F\vec{k}}^{\left(-\right)}$ or equivalently $\Psi_{FElm}^{\left(-\right)}$, which is independent of the former, as eigenvector of the molecular hamiltonian already considered.

Projection on free hamiltonian eigenstates amount to wavepacket analysis, giving the corresponding transition probabilities.

1.In the multiphoton domain, generally the lowest order perturbation theory is employed
(30)$$\Phi_{DI}^{Nph}=D{\left(H-E_{N-1}\right)}^{-1}D{\left(H-E_{N-2}\right)}^{-1}D\cdots D\Phi_{I}=D{\left(H-E_{N-1}\right)}^{-1}\Phi_{DI}^{Nph-1}$$
$$E_{m}=E_{i}+m\hbar \omega$$
valid for a single photon field, and non-resonant intermediate states, but can be easily generalized, and can be evaluated recursively.2.For nonperturbative fields a standard approach is now the solution of the time dependent Schrödinger equation (TDSE)
(31)$$\Phi_{DI}\left(t\right)=U\left(t\right)\Psi_{I}\;\;\;\;\;-i\frac{d}{dt}\Phi_{DI}\left(t\right)=H\left(t\right)\Phi_{DI}\left(t\right)$$
with a time dependent hamiltonian which includes the external field.

Both approaches have become rather standard for computational simulations of the relevant processes. As the time evolution is unitary, at all times the wavepacket is square integrable, and can be accurately described by the B-spline basis as long as its amplitude at the boundary remains negligible. This may require very long range bases, spanning hundreds or even thousands of atomic units [25], which can, however, be easily accomodated in the B-spline basis. Techniques like the use of complex absorbing potentials at the boundary can reduce the need of such extended ranges [26].

One typical approach is employing a spectral method, i.e., a preliminary full diagonalization of the free molecular hamiltonian,
$$H\Psi_{I}=E_{I}\Psi_{I}$$
so that the resolvent can be expanded as
(32)$${\left(H-E\right)}^{-1}=\sum_{I}\frac{|\Psi_{I}\rangle \langle \Psi_{I}|}{E_{I}-E}$$
and the amplitude (Equation 30) can be easily evaluated. Similarly, the TDSE equation, after expansion of the wavepacket
(33)$$\Phi_{DI}\left(t\right)=\sum_{I}C_{I}\left(t\right)\Psi_{I}$$
is reduced to a well conditioned system of ordinary differential equations
(34)$$-i\frac{dC_{I}\left(t\right)}{dt}=\sum_{J}\left(E_{I}\delta_{IJ}+V{\left(t\right)}_{IJ}\right)C_{J}\left(t\right)$$
which can be solved by a variety of approaches.

In the static-DFT framework one is working with an independent particle hamiltonian
(35)$$H=h_{KS}\left(1\right)+\cdots +h_{KS}\left(N\right)$$
(36)$$h_{KS}\phi_{i}=\varepsilon_{i}\phi_{i}\;\;\Rightarrow H\Phi_{I}=E_{I}\Phi_{I}$$
(37)$$\Phi_{I}=|\phi_{I_{1}}\cdots \phi_{I_{N}}\rangle \;\;\;\;\;\;E_{I}=\varepsilon_{I_{1}}+\cdots +\varepsilon_{I_{N}}$$

The single particle $h_{KS}$ can be fully diagonalized even with large B-spline bases. Then all equations and matrix elements reduce to one-particle problems and large systems become affordable. While this is the most basic level, as it lacks correlation among the states, it is capable of giving a generally semiquantitative description [27]. Note that, as the coupling with the external electromagnetic field, $h\left(t\right)=h_{KS}+V\left(t\right)$, is a single particle operator, within this approximation the full hamiltonian remains of single particle type, and it is easy to show that the full *N*-particle TDSE is equivalent to the solution of individual one-particle TDSE for the initial orbitals
(38)$$-i\frac{d}{dt}\phi_{i}\left(t\right)=h\left(t\right)\phi_{i}\left(t\right),\;\;\;\;\Phi \left(t\right)=|\phi_{1}\left(t\right)\cdots \phi_{N}\left(t\right)\rangle \;\;\Rightarrow \;\;-i\frac{d\Phi (t)}{dt}=H\left(t\right)\Phi \left(t\right).$$

This means that the single particle states $\phi_{i}$ are propagated independently, without any restriction, and due to the unitarity of the evolution they remain orthonormal at all times. Moreover, as the exact solution of a model hamiltonian, the result is gauge independent, provided the basis is sufficiently accurate, as has been well verified numerically.

Final projection of the wavepacket $\Phi_{DI}$, either from multiphoton or time propagation, is straightforward, with the corresponding continuum functions already available in the same B-spline basis. In other formulations, techniques based on the evaluation of the electronic flux at the boundary can be employed [26].

It may be remarked that most algorithms reduce to straightforward operations of linear algebra that can be parallelized efficiently also with the use of optimized libraries and scale very well.

## 4. Ab Initio Developments

The B-spline basis maintain excellent performance in an ab initio setting [28]. Actually, most algorithms employed remain unchanged, the main difference being the choice of the hamiltonian, or implicitly the structure of the many electron basis, and the evaluation of matrix elements. An obvious choice is the close-coupling approach, but other structures like multiconfigurational SCF can be equally employed.

Clearly, a prerequisite is the availability of two electron integrals, relative to the Coulomb interaction, in the LCAO basis
(39)$$\langle \phi_{i}\left(1\right)\phi_{j}\left(2\right)|\phi_{k}\left(1\right)\phi_{l}\left(2\right)\rangle =\int \frac{\rho_{ik}\left(1\right)\rho_{jl}\left(2\right)}{r_{12}}\;dr_{1}dr_{2}$$
relative to the electron densities $\rho_{ik}\left(1\right)=\phi_{i}\left(1\right)\phi_{k}\left(1\right)$ and similarly for $\rho_{jl}\left(2\right)$. The one-center radial two electron integrals have been long available [29,30], and the angular part is analytical, as is well known in atomic theory. Actually, for not too large systems, that may be a reasonable compromise, as it is partly used in some current well known codes [2,3]. The full evaluation of LCAO B-spline two electron integrals, essentially by numerical approaches, is demanding but by no means unfeasible. One straightforward approach is full numerical integration. Due to the limited range of functions on atomic centres, all integrals will have one density (particle 1, ρ(1)) restricted to an atomic sphere, whose Coulomb potential may be easily evaluated in multipolar form and integrated over the second particle density ρ(2), which in many cases is also restricted to an atomic sphere.

A second approach is to solve the Poisson equation for the first density. As the basis is very accurate, so is the computed potential. It requires implementation of multipolar boundary conditions at $R_{max}$, which is easily implemented with B-splines. After the evaluation of the the expansion of the density in the basis, $\rho_{\mu 1}=\langle \chi_{\mu }|\rho_{1}\left(1\right)\rangle$ the corresponding potential
(40)$$V_{1}=\sum_{\mu }v_{\mu 1}\chi_{\mu }$$
is obtained by solving the linear system
(41)$$\sum_{\mu }\Delta_{\nu \mu }v_{\mu 1}=-4\pi \;\rho_{\nu 1}$$

$V_{1}$ is then integrated with the second density $\rho_{2}$
(42)$$\int \rho_{1}\left(1\right)\frac{1}{r_{12}}\rho_{2}\left(2\right)\;dr_{1}dr_{2}=\int \rho_{2}\left(1\right)V_{1}\left(1\right)\;dr_{1}=\sum_{\mu }v_{\mu 1}\rho_{\mu 2}$$
which reduces to the last equation if $\rho_{\mu 2}=\langle \chi_{\mu }|\rho_{2}\rangle$ is also available. This approach is particularly effective if not all integrals over primitive $\chi_{\mu }$ functions are required. In a close coupling expansion only integrals of the type
$$\langle \chi_{\mu }\phi_{i}|\chi_{\nu }\phi_{j}\rangle ,\;\;\;\;\langle \chi_{\mu }\chi_{\nu }|\phi_{i}\phi_{j}\rangle ,\;\;\;\;\langle \chi_{\mu }\phi_{i}|\phi_{j}\phi_{k}\rangle$$
are required and the relative smooth $\rho_{1}$ densities are accurately approximated in the B-spline basis. Moreover, this avoids the need for integral transformation from primitive to the MO basis.

A general routine for these two electron integrals has been recently completed and will be separately reported with full details [31]. It is used in the B-spline ADC(2) code developed by Ruberti [32], mainly devoted to strong field and attosecond processes.

The second main point is the evaluation of the matrix element in the chosen *N*-particle basis, including dealing with nonorthonormality. In the case of close-coupling expansions, explicit formulas entail up to three particle transition density matrices between the bound states $\left\{\Psi_{I}^{N-1},\Psi_{K}^{N}\right\}$ [7,9]. Some implementations have been discussed in the literature, and represent generally a computationally expensive task, although special algorithms may afford significant speedup.

Explicit formulas have been derived for the lowest level, CIS wavefunctions, and have been implemented at the OCE level [28]. Implementation with RASCI wavefunctions is currently studied. These will be available in a future release of Tiresia. Moreover, separate treatment of Auger and resonant Auger decay [33] via golden rule formulation is currently underway.

## 5. Applications

### 5.1. Molecular Photoionization

The most common application is the description of single photon molecular photoionization. This is interesting per se, as it is an important probe of molecular electronic structure, or as an initial step of strong field or ultrafast phenomena, as well as a probe in time dependent studies. From the transition amplitudes and K-matrix the cross section observables σ and β, which define the differential cross section for randomly oriented molecules, and linear polarization
(43)$$\frac{d\sigma }{d\theta }=\frac{\sigma }{4\pi }\left[1+\beta_{2}P_{2}\left(\cos\theta \right)\right]$$
can be obtained. $P_{2}\left(\cos\theta \right)$ is the second Legendre polynomial and θ is the angle between light polarization and photoelectron momentum. More complex angular distribution, for partially aligned or oriented molecules, nondipole contributions, etc, can also be obtained.

Photoionization of C${}_{60}$ has been investigated several times [34,35,36,37]. Illustrative results from recent studies are reported in Figure 2. A comparison of DFT and TDDFT results show a very similar shape of the total cross section, with a significant increase of the absolute value in TDDFT (left upper panel). Note that the latter incorporates oscillator strength due to discrete excitations above threshold that do not appear in the DFT cross section but show up as autoionization resonances in TDDFT. Actually the fixed nuclei cross section shows an amazing array of very sharp structures, arising both from shape and autoionization resonances. These are washed away by nuclear motion, and we have actually broadened the computed profiles for a better comparison with the experiment, which is reproduced quite accurately. To show, however, the high resolution attainable, a very fine energy scan of the total DFT core cross section is reported in the central upper panel for the first 20 eV KE. Considering that only shape resonances can appear at the DFT level, the presence of these sharp structures is very surprising. Clearly a specific feature of C${}_{60}$, it hints at the presence of very localized quasi bound states in the one-particle continuum. The absence of dramatic effects in going from DFT to TDDFT results appears to disprove the claim of giant plasmonic effects in C${}_{60}$, which derives from results obtained with jellium models, which smear the carbon cores over a spherical shell. Indeed, seen on a large energy scale, total valence and core ionization of C${}_{60}$ follows closely the sum of 60 free-atom cross sections [37]. TDDFT becomes very close to DFT some 150 eV above threshold. Indeed the jellium models, lacking the hard cores which are responsible for the high energy cross section, give results that decay much too fast with increasing KE. Still, a lot of structure survives up to pretty high energies, as strong oscillations in the cross section. These are best seen in individual ionizations, like the highest occupied molecular orbital (HOMO) 6H${}_{u}$ channel (left lower panel). The phase of oscillation is opposite for states of opposite parity, so that their cross section ratio enhances the oscillations (right lower panel). These have been detected since the first photoemission studies in C${}_{60}$. The DFT results match closely the experimental data available up to about 300 eV [38]. Further results suggest that the oscillations persist undamped up to at least 600 eV, after which they become irregularly damped [37]. However, nuclear motion neglected in this study may induce some earlier damping.

Oscillations in the cross sections are indeed quite ubiquitous, caused by diffraction effects due to multicenter emission of the electron wave [39,40,41,42]. Further structures at high energies are associated with non-dipole effects [43].

Correlation effects in bound states due to ionization have been noticed and explained long ago [44,45], especially as concerns ionization energies and the appearance of satellite states. One subtle manifestation is a modification of the orbital structure in the ion. In most small molecules the orbital structure is quite rigid, and ground state orbitals, either HF or DFT, change very little in the ion. However, in larger systems, a so called hole-mixing effect, associated with a rotation of the GS orbitals in the ion, can be relevant [19,46]. An example is in low symmetry molecules, where many MOs can remix without symmetry constrains. Another typical situation is in transition metal compounds, where the mixing between metal d and ligand orbitals can change dramatically by electron removal. Whie often signalled by large changes in ionization energies with respect to Koopman’s values, the orbital change, which is embodied in the corresponding Dyson orbital, can be detected only by transition properties, like photoionization observables or electron momentum profiles [47]. A recent study on a prototypical system, bis-allyl nickel Ni(C_3_H_5_)_2_ [19], shows the large changes in σ and β parameters from those afforded by HF or DFT orbitals (Figure 3). Moreover, it shows the value of photoionization observables concerning aiding in the difficult problem of assigning the spectrum in strongly correlated systems, e.g., the appearance of the strong 3p→3d autoionization resonance in ionizations involving metal 3d participation.

A new parameter $\beta_{1}$ appears in the angular distribution of photoelectrons in photoionization of chiral molecules with circularly polarized light:(44)$$\frac{d\sigma }{d\theta }=\frac{\sigma }{4\pi }\left[1+m_{r}\beta_{1}\cos\theta -\frac{1}{2}\beta_{2}P_{2}\left(\cos\theta \right)\right]$$
where $m_{r}$ is +1 or −1 for left and right circular polarization. This can be measured and is an important tool for the study of molecular chirality. It arises purely from an electric dipole transition, and the effect is orders of magnitude stronger than circular dichroism, which is very difficult to measure for diluted samples in gas phase. DFT calculations have often afforded a fair reproduction of the experimental results [48]. However, the effect is largest close to the threshold, and dies after about 50 eV KE. Unfortunately the threshold region is the most difficult to describe, and is very sensitive to the specific DFT potential employed, as illustrated in the case of camphor HOMO ionization in Figure 4. The left upper panel taken from [49] shows results from a multiple scattering X${}_{\alpha }$ calculation and present LB94 results, compared to experiments. Other panels show results obtained with additional choices of the potential, employing GS or transition state densities. A further problem is the proper treatment of correlation effects, e.g., in the case of chiral transition metal molecules, which are still pretty large for higher level ab initio approaches.

A case is photoionization of Cobalt tris-acetylacetonate. The photoelectron spectrum, with superimposed KS eigenvalues from the LB94 potential, and IEs calculated with the outer valence greens function (OVGF) approach [44], a usually pretty accurate treatment for ordinary organic molecules, is presented in Figure 5. While HOMO ionization is reasonably described by DFT, the following ones are clearly inadequate. As a consequence the computed $\beta_{1}$ parameter is in fair agreement with the experiment for HOMO ionization, but pretty off for the following ones [50,51] (Figure 6).

All results considered are evaluated at a fixed nuclear geometry. To include nuclear motion effects it is necessary to evaluate transition dipoles at various geometries and integrate over vibrational wavefunctions [40,41]. Only one-dimensional problems have been considered, but it is easy to generalize to multidimensional models currently available. Vibronic coupling effects have been amply observed [40,41,42]. More demanding is the simulation of pump-probe femtosecond experiments via time resolved photoelectron spectra (TRPES) that require evaluation of photoionization observables for hundreds or more nuclear configurations. For this reason up to now only rather crude approximations, like plane or coulomb waves, have been considered, and computational efficiency becomes of paramount importance. A recent simulation of pyrazine employing a surface-hopping method and the Tiresia code has been presented [52].

A comparison of experimental and calculated time resolved spectra is presented in Figure 7. The agreement is quite satisfactory, some deficiency at the lowest KE is probably due to inaccuracy of the cross section close to threshold. Up to now laser ionization has been employed almost exclusively in this field, and photon energy available is quite low. It is expected that such experimental limitations will be soon overcome. In this respect the simulation may even suggest an optimal photon range for maximum sensitivity to the dynamical aspects investigated. Moreover, additional information may be gained if molecular orientation is not random, but prealigned with an additional laser pulse, so that a richer angular distribution is achieved.

### 5.2. Strong Field and Ultrafast Processes

The program has been used rather extensively to describe and analyze wavepackets in strong field ionization and attosecond pump-probe experiments [53,54] (see also the program by Ruberti [32] based on the same B-spline basis). One of the first applications was to study the angular dependence of strong field ionization probability, and to highlight the role played by ionization of deeper levels besides HOMO [55]. Some results of a study of water [56] are reported in Figure 8. It illustrates the dependence of ionization yield on the laser intensity (at λ=800 nm). The inset shows how the simple MO-ADK approach that was often used, based on the exponential dependence on ionization energy, overestimates the HOMO contribution except at the highest intensities. Moreover, the different angular dependence of the ionization yield on the angle with the laser polarization, for different orbitals, may even switch the dominant contribution. These results are obtained at the DFT level, and it has been recently shown that important modifications are induced by interchannel coupling [57].

The full energy resolved photoelectron spectrum relative to HOMO ionization, with laser polarization along *Z* is reported in Figure 9 for different laser intensities. It is interesting to see as, starting with a regular series of peaks corresponding to individual photon absorption, the spectrum becomes chaotic at the highest intensities.

Finally, we present a calculation of HHG emission in CO${}_{2}$ at different field intensities (Figure 10). It was experimentally found that a minimum in the HHG emission appears, shifting to higher harmonics as intensity is increased. The DFT results reproduce the trend, although they underestimate the harmonic order of the minimum. A more refined calculation, which included interchannel coupling at essentially the CIS level, is able to obtain complete agreement [58]. It is clear that a lot of information and insight can be obtained already at the DFT level, as might be expected from the great success of one-electron models to explain most of the strong field phenomena. For quantitative agreement, however, some level of correlation has to be included, even at the lowest level, like CIS of linear response TDDFT. This is important since higher-level correlated approaches become often prohibitively expensive in larger systems.

## 6. Conclusions

The structure of the Tiresia code for describing continuum states as well as electronic wavepackets has been presented. The multicenter (LCAO) B-spline basis set has been illustrated in detail. It affords:A dense set within a finite range (a sphere) of arbitrary length. It can approach completeness and therefore converge to the required solutions.A complete control of the overlap matrix, hence numerical linear independence and stability.Accurate solutions of homogeneous and inhomogeneous equations within the range, with proper boundary conditions, which are easily implemented.

In particular for photoionization studies, it affords convergent solutions of the multichannel problem, at selected energies from threshold up to very high KE (10 keV have been reached) and arbitrary energy resolution, for ionizations from valence to deep core, and complex molecular systems. In the present static-DFT and TDDFT formulations, as well as Dyson-DFT and Dyson-TDDFT, it provides a rather accurate description of a vast phenomenology. The algorithms implemented are simple, mostly based on linear algebra, and easily parallelized, leading to efficient computation that makes the treatment of complex systems, and calculations of many geometries easily affordable. Important steps towards the implementation of fully ab initio close coupling formulations have been performed, and further work is in progress. The same basis and hamiltonians allow calculation of multiphoton and strong field processes, in particular time propagation of wavepackets in external electromagnetic fields, and their analysis by projection on free field states of the same hamiltonian. A number of examples from past results and current work are provided for illustration.

## Figures and Tables

**Figure 1 molecules-27-02026-f001:**
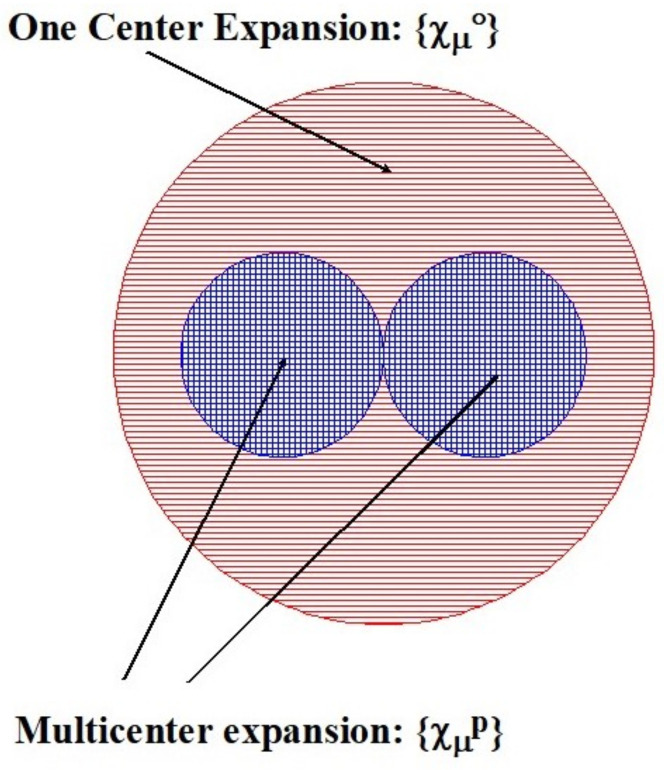
Multicenter (LCAO) expansion.

**Figure 2 molecules-27-02026-f002:**
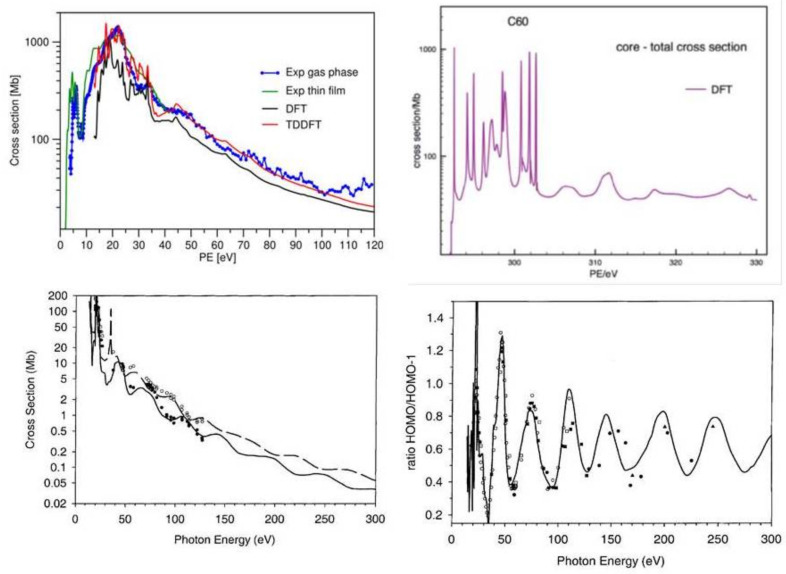
C${}_{60}$ photoionization. Upper left: Total DFT and TDDFT cross section, from Ref. [36] with permission. Upper right: High resolution DFT core cross section [37]. Lower left: HOMO and HOMO-1 cross sections; lower right HOMO/HOMO-1 cross section ratio (circles are experimental data), from Ref. [38] with permission.

**Figure 3 molecules-27-02026-f003:**
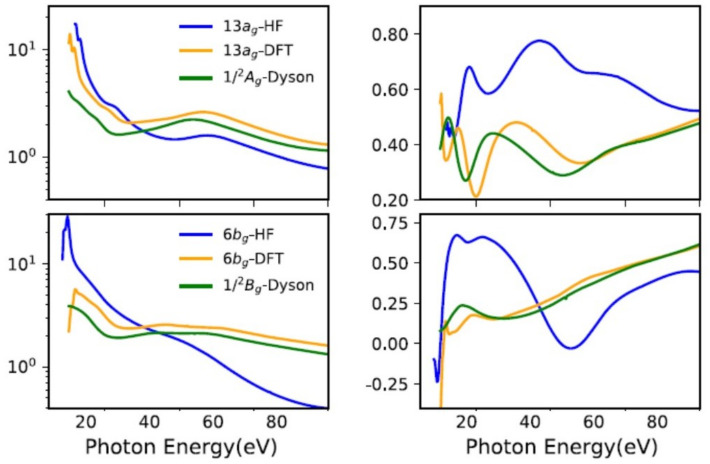
Nickel bis-allyl cross section and β parameter at the DFT, HF and Dyson-DFT level. From Ref. [20] with permission.

**Figure 4 molecules-27-02026-f004:**
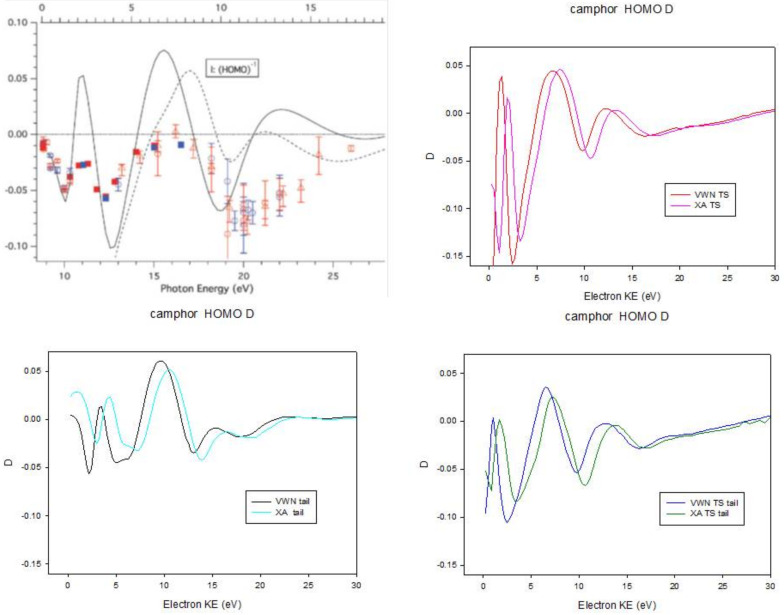
Dichroic parameter $\beta_{1}$ for Camphor HOMO ionization. Results for different potential choices. Left upper panel from ref. [49] with permission.

**Figure 5 molecules-27-02026-f005:**
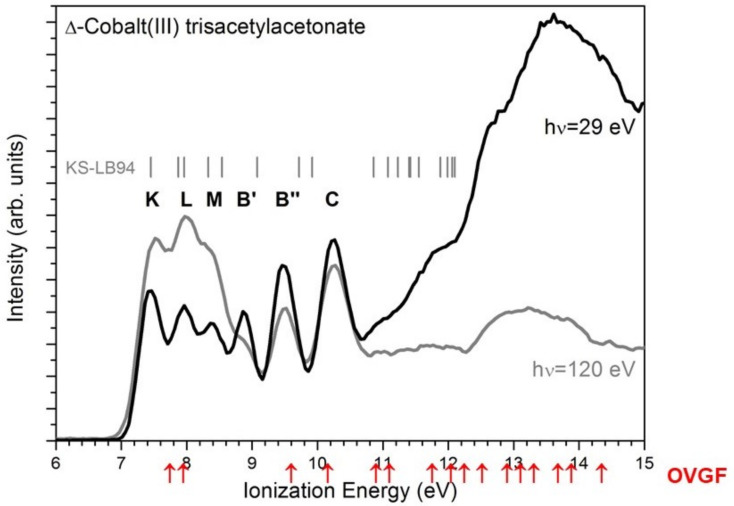
Valence photoelectron spectrum of cobalt tris-acetylacetonate, with LB94 eigenvalues (black) and OVGF IEs (red). Experimental spectrum from [51] with permission.

**Figure 6 molecules-27-02026-f006:**
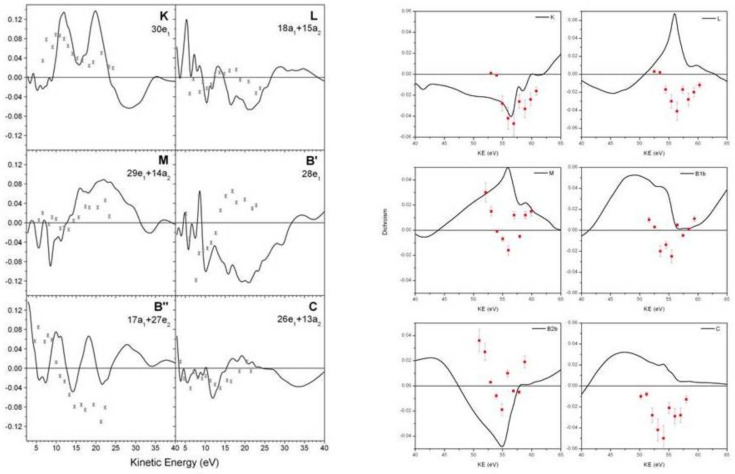
$\beta_{1}$ parameter computed for the valence levels of Cobalt tris-acetylacetonate (from [51] with permission). Right panel in the region of autoionization resonance.

**Figure 7 molecules-27-02026-f007:**
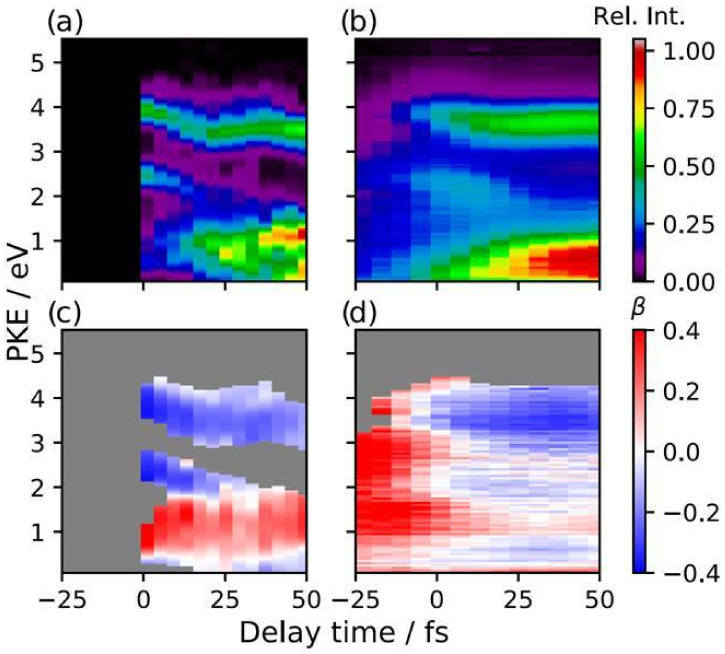
Time resolved photoelectron spectra of pyrazine. Cross section (**upper** panels), β parameter (**lower** panel). (**a**,**c**) experimental; (**b**,**d**) calculated. From [52] with permission.

**Figure 8 molecules-27-02026-f008:**
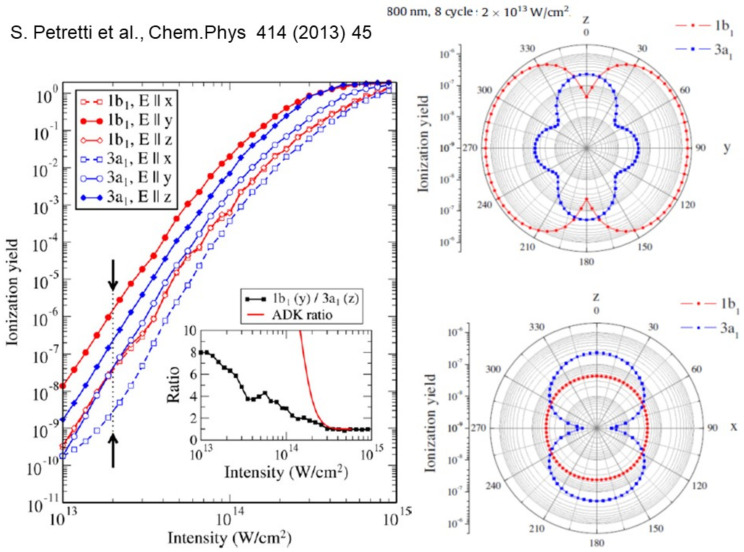
Strong field ionization yield in water (from [56] with permission).

**Figure 9 molecules-27-02026-f009:**
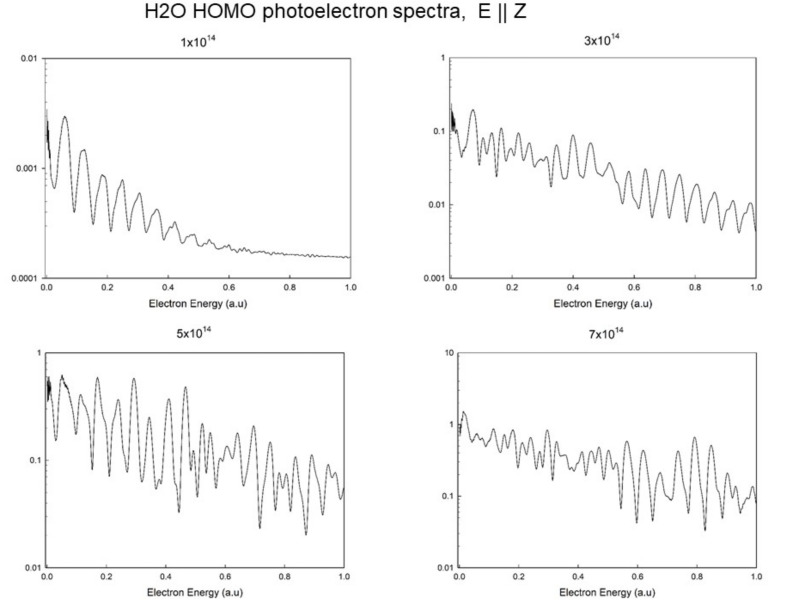
Photoelectron spectrum (ionization probability) relative to HOMO ionization in water at $I=1.0\times 10^{14}$, $3.0\times 10^{14}$, $5.0\times 10^{14}$ and $7.0\times 10^{14}$ (blue) W/cm${}^{2}$.

**Figure 10 molecules-27-02026-f010:**
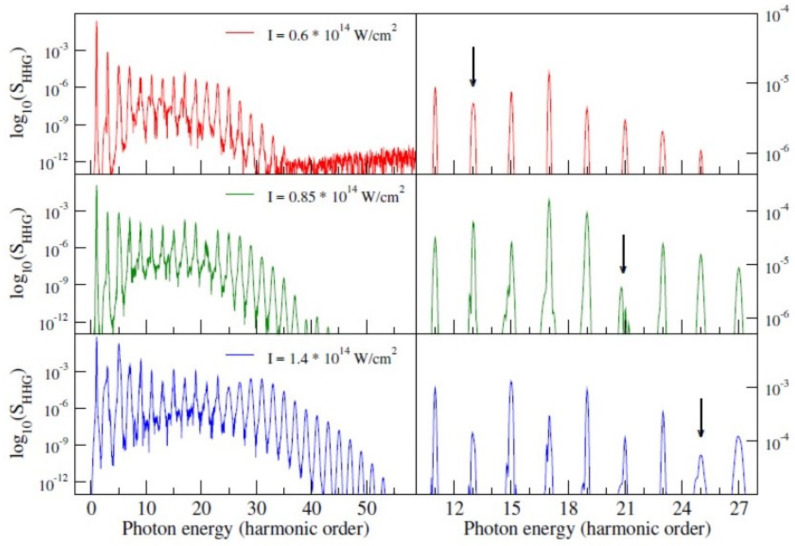
HHG spectra of $CO_{2}$ at $I=0.6\times 10^{14}$ (red), $0.85\times 10^{14}$ (green) and $1.4\times 10^{14}$ (blue) W/cm${}^{2}$.

## Data Availability

Data available from the authors upon request.

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
