# Peer review of "Continuum Electronic States: The Tiresia Code"

_molecules, 2022, doi:10.3390/molecules27062026_

Round 1
Reviewer 1 Report
I recommend the paper "Continuum Electronic States: the Tiresia Code" for publication in the journal "Molecules" with the corrections recommended in the attached PDF file.

Author Response
We thank the referee for the constructive criticism.
Almost all suggestions have been taken into account, the rather small changes in the manuscript are detailed below
- In response to the query about eq. (7), we have taken the opportunity to give more detail about the basis (new equations (7) and (8), to define ‘c’ in eq. (11).
- Static exchange(SE) has been defined at the start of section 2.3.1
- Misprints in old eq. (18), now (22) have been corrected. It was hardly comprehensible because current template does not support Latex \mathbf, giving wrong symbols. For the rest the presentation is identical as found in paper MCD21
- The phrase involving LB94 (present line 211) hase been reworded according to suggestion
- LOPT has been removed
- The notation for the two-electron integrals has been explained (new eq. (39))
- Sigma and beta (just before eq. (43)) have been reworded according to suggestion (and now called observables)
- HOMO has been spelled out (line 322)
- OVGF has been spelled out (line 304)
- The figures have been sharpened and enlarged, and captions have been worked out in more detail.

Reviewer 2 Report
In this paper, the authors introduce the Tiresia code for continuum electronic states calculations. The structure of the Tiresia code for describing continuum electronic states and electronic wavepackets are presented and discussed. The multicenter (LCAO) B-spline basis set is also discussed in detail. Some examples from previous results and current work are illustrated. This paper is interesting in the point view of the introduction of Tiresia program for first-principles continuum states calculations. The manuscript is well presented and clearly written.
This paper should be published after a minor improvement as follows.
- Labels for figure 2, 4, 6, 8 and 9 are too small. Please improve the figures.
Author Response
Thank you for the suggestion. We enlarged and uploaded some new figures
Reviewer 3 Report
Report on the paper entitled "Continuum Electronic States:the Tiresia Code", by Piero Decleva, Mauro Stener, and Daniele Toffoli, submitted to Molecules.
The present paper describes a multicenter B-spline basis code providing convergent solutions for electronic continuum states and wavepacket propagation. The code delivers efficiently accurate description of photoionization properties of complex systems, both in the single photon and strong field environments. Several examples are provided.
The paper is well written, and describes satisfactorily the work. I have a few suggestions to be considered, but they are not mandatory:
- A comparison with the possibilities of the Wien2k code would be welcome.
- One of the oldest calculation on C60 with a comparable spatial partition of the density was given in Int. J. Quant. Chem. 1994, 51, 319. A comparison of the MOs diagram could be made.
I recommend publication in the present form.
Typo: thre
Author Response
Thank you fo the suggestions
I apologize, I'm aware but not at all familiar with the Wien2k code. I see that it is a planewave code for periodic solids, so with a scope pretty different from molecular applications, and I do not feel confident to cite properly.
Also I have no access to the old IJQC paper, very many calculations of the GS are available in the literature, prior to the present work. We focused on our own not to discuss C60 in detail, but as examples of the capabilities